

# Successful treatment of diplopia using prism correction combined with vision therapy/orthoptics improves health-related quality of life

Juan Antonio Portela-Camino[1], Irene Sanchez[2,3], Cristina Gutierrez[1] and Santiago Martín-González[4]

[1] Department of Optometry, Begira Ophthalmologic Clinic, Bilbao, Spain
[2] Departamento de Física Teórica Atómica y Óptica, Universidad de Valladolid, Valladolid, Spain
[3] Instituto de Oftalmobiología Aplicada (IOBA), Universidad de Valladolid, Valladolid, Spain
[4] Department of Construction and Manufacturing Engineering, University of Oviedo, Gijon, Spain

Corresponding author
Irene Sanchez,
isanchezp@ioba.med.uva.es

## ABSTRACT

**Background:** To track improvement in diplopia symptoms with strabismus-specific health-related quality of life (HRQOL) questionnaire across a treatment consisting of prism correction followed by vision therapy/orthoptics when prism treatment alone has not succeeded.

**Methods:** Forty-eight participants with diplopia and a mean age of 62.45 were asked to complete an Adult Strabismus-20 (AS-20) questionnaire and a Diplopia Questionnaire (DQ) before and after prism correction. Inclusion criteria were diplopia reported on the DQ as "sometimes", "often" or "always" at reading or straight-ahead distance. The prism correction was classified as successful if the participant reported "never" or "rarely" on the DQ for reading and straight-ahead distance; and unsuccessful if the perceived diplopia worsened or remained the same. For all participants, mean initial AS-20 scores were compared with mean post-prism correction scores, taking into account AS-20 subscales (reading and general functions, and self-perception and interaction). Participants in the failed prism treatment subgroup subsequently underwent a programme of vision therapy wearing their prism correction, the results of which were again determined by participants' responses on the AS-20 questionnaire, completed before and after the vision therapy.

**Results:** Five of the 48 participants dropped out of the study. Prism correction was classified as successful in 22 of 43 participants (51%), and unsuccessful in 21 (49%). Those participants for whom the prism correction was classified as a success showed a statistically significant improvement ($p = 0.01$) in both reading and general functions. In the failed treatment subgroup, no significant change in AS-20 score was recorded for any of the domains ($p = 0.1$). After treatment with vision therapy/orthoptics, however, 13 of the 20 participants in the unsuccessful prism correction subgroup (one of them dropped out the study) achieved binocular vision and statistically significant improvement in reading and general functions ($p = 0.01$).

**Conclusions:** Although effective prism correction of diplopia is correlated with enhanced HRQOL, prism correction alone is frequently not sufficient to achieve this objective. In these cases, vision therapy/orthoptics treatment as a coadjutant to prism correction is shown to improve HRQOL.

## INTRODUCTION

Globally, the prevalence of strabismus ranges from 2% to 4% (*Fieß et al., 2020*). Diplopia in adulthood is associated with strabismus (*Philipps, 2017*) deriving from different aetiologies: decompensation of previous deviations (phoria or tropia) (*Cioplean & Nitescu Raluca, 2016*; *Wang et al., 2019*); paresis or paralysis of extraocular muscles, whether vascular (*Watanabe et al., 1990*; *Patel et al., 2005*), tumour-related (*Sharma et al., 2010*) or secondary to brain trauma (*Kaido et al., 2006*); autoimmune diseases, such as Graves' ophthalmopathy (*Johnson, Jameyfield & Aakalu, 2021*); or problems secondary to retinal diseases, such as dragged-fovea diplopia syndrome (*De Pool et al., 2005*).

The impact of strabismus on health-related quality of life (HRQOL) in adults can be assessed using different questionnaires such as the specific Adult Strabismus-20 questionnaire (AS-20) (*Hatt et al., 2009a*), or the more general American National Eye Institute Visual Functions Questionnaire (VFQ-25) (*Mangione et al., 2001*). The AS-20 has been shown to be more sensitive than the VFQ-25 for detecting reduced HRQOL in adult strabismus (*Hatt et al., 2009b*). A previous evaluation of the psychometric properties of the AS-20 with Rasch analysis proposed the reduction of the questionnaire to four distinct domains: self-perception (five items), interactions (five items), reading function (four items), and general function (four items) (*Leske et al., 2012*).

The impact of diplopia on quality of life has been analysed by different authors and these data have been put together in a systematic review (*McBain et al., 2014*). *Holmes et al. (2013)* designed the Diplopia Questionnaire (DQ) specifically to quantify diplopia, allowing the position and distance at which the patient experiences diplopia to be recorded. This research has proven a high correlation between the functional results of the AS-20 and measurements provided by the DQ in patients with diplopia in primary gaze and reading (*Holmes et al., 2013*).

Regarding the effects of different strabismus treatments on HRQOL, a number of studies propose surgery as an option associated with significant and enduring functional and psychological improvement in patients with strabismus (*Holmes, Liebermann & Hatt, 2016*; *Holmes, Hatt & Leske, 2009*; *Hatt et al., 2012*). Moreover, several studies have indicated that improvements in HRQOL are greater in diplopia patients than those without diplopia (*Holmes, Liebermann & Hatt, 2016*; *Holmes, Hatt & Leske, 2009*; *Hatt et al., 2012*; *Sim et al., 2018*).

*Hatt et al. (2014)* studied the effects of prism correction in participants with binocular diplopia, achieving a 68% success, rate measured with the DQ test, taking account participants with diplopia rarely or never perceived in primary gaze and reading. Improvement was also recorded in the reading and general functions subcategories of the AS-20, but did not extend to either the self-perception or interactions categories (*Hatt et al., 2014*).

Vision therapy/orthoptics has proven its effectiveness in improving vergence ranges in patients with esotropia (*Molina-Martín et al., 2020*) and convergence insufficiency

(*Scheiman et al., 2020*); to the best of our knowledge, however, the effects on HRQOL of this treatment have never been evaluated. The aim of the present study is to assess the impact on the HRQOL of adults with diplopia of a protocol that involves prism correction and vision therapy/orthoptics as required.

## MATERIALS AND METHODS

### Design

A pre-post pseudo-experimental study design was used. Participants were adults (>18 years old) with acquired diplopia (duration >3 months), whose responses to the DQ indicated episodes of diplopia at near or far distance at least 50% of the time. Patients with a history of previous treatment (prism correction, vision therapy, strabismus surgery or botulinum toxin) were also included.

Participants were recruited among the patients of two centres: Ikusgune Optometric Centre and Begira Ophthalmologic Clinic, Basque Country (Spain), specifically patients with strabismus who were able to achieve fusion at the objective angle with the synoptophore. Patients presenting with complete paralysis of the 3rd, 4th, or 6th cranial nerves (*i.e.*, no duction during occlusion of the dominant eye), severe monocular or binocular amblyopia (best corrected visual acuity < 0.2), monocular diplopia, nystagmus, or mental or cognitive impairments that would rule out the use of a HRQOL assessment, were excluded.

The study followed the tenets of the Declaration of Helsinki and was approved by the Basque Country Ethics Committee of Clinical Research (PI2021059). The participants signed a consent form after receiving a verbal and written explanation of the study.

### Evaluation protocol

#### HRQOL assessments

The DQ provides a self-evaluation of diplopia severity on a five-point scale (never, rarely, sometimes, often, always) in seven gaze positions (reading, straight-ahead distance, right, left, up, down, and other gaze position). According to the authors' criteria (*Holmes et al., 2013*), responses indicating diplopia perceived by the participant "sometimes", "often" or "always" in reading and straight-ahead distance gaze (primary gaze at far) should be included in the analysis. Successful prism correction was defined as diplopia rated on the DQ as "never" or "rarely" in reading and straight-ahead distance gaze positions.
The validated version of the AS-20 questionnaire was also used (*Leske et al., 2012*). Each of the four domains (self-perception, interactions, reading function, and general function) was scored independently and finally consolidated into a unique 0 to 100 score (worst to best HRQOL) to facilitate interpretation. Participants completed the AS-20 questionnaire and the DQ before and after prism correction, and again after vision therapy/orthoptics as applicable.

#### Clinical evaluation and follow-up visits

An experienced optometrist evaluated the participants in an initial clinical evaluation. Refractive error was corrected, and best corrected visual acuity (BCVA) was obtained using

the HOTV visual acuity chart with crowding bars (Smart4Vision, Barcelona, Spain). Binocular vision was analysed using the Worth Four Dot test, at a distance of four metres with scotopic illumination (noting a numerical value of five for diplopia; four for fusion; and two and three in cases of suppression), and the Random Dot Preschool Stereoacuity test (Stereo Optical Company Inc., Chicago, IL, USA) at near distance, following the manufacturer's instructions. Patients whose responses indicated nil stereoacuity an arbitrary value of 1,300" (ecological stereoblindness) was assigned (*Chopin, Bavelier & Levi, 2019*).

Deviation angle was measured using two different procedures. The first of these was a cover test with an accommodative stimulus, at near and far distances, with the prism placed before the strabismic eye, using stimuli based on characters two lines below the participant's BCVA. If the participant's strabismus combined horizontal and vertical deviations, the primary deviation was corrected first (*e.g.*, the vertical deviation in 4th cranial nerve paresis). The second procedure involved the use of a synoptophore. Objective deviations were evaluated (to test whether the patient was able to fuse at the objective angle) using a traditional synoptophore (Oculus, Wetzlar, Germany), and a modern version based on virtual reality glasses with an eye tracker (VisionaryVR; VisionaryTool S. L., Gijón, Spain). Deviations in cyclotorsion were measured with both synoptophore devices in addition to a Double Maddox rod test (*Liebermann et al., 2021*).

The clinical evaluation included an ocular health exam (using tropicamide for pupil dilation), biomicroscopy, an indirect ophthalmoscope, and macular optical coherence tomography (Maestro2; Topcon, Tokyo, Japan).

At follow-up visits on completion of both prism correction and vision therapy/orthoptics treatment phases, binocular vision was retested with the Worth Four Dot and the Random Dot Preschool Stereoacuity test, and participants were required to complete both AS-20 and DQ questionnaires.

## Treatment protocol

Treatment lasted from 1 to 4 months, with two clearly differentiated phases (Fig. 1).

### *Phase I. Prism correction*

At the first visit, prism correction was performed. Prisms were prescribed, generally using the minimum amount of prism needed to eliminate the participant's diplopia in straight-ahead distance, reading or both gaze positions. The following methodology was adhered to:

i) the first prism correction value (the first prism with which no movement in strabismic eye was observed) was obtained by unilateral cover test, with the participant fixating on the 20/400 letter "E" on the Snellen chart at far or near distance (*Tamhankar, Ying & Volpe, 2012b*);

ii) in cases of horizontal and vertical deviation, tables were used to obtain the angle and hence the power at which to prescribe the equivalent oblique prism (*Reinecke et al., 1977, 2001*);

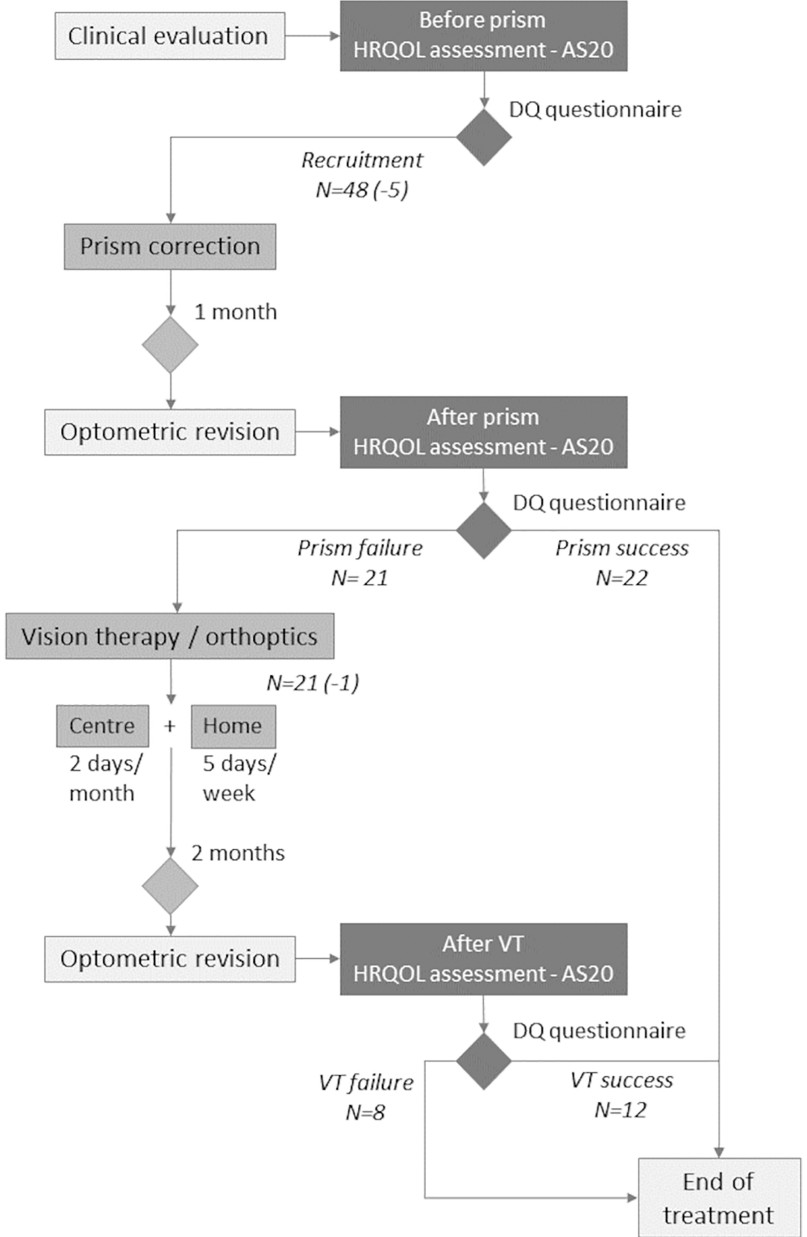

**Figure 1 Flowchart of Methods section.** Flowchart of proposed treatment of prism correction followed by orthoptic/vision therapy. The number of volunteers at each step includes the number of abandons or new inclusions in parentheses.               

iii) the prism was placed before either the dominant or the strabismic eye, according to the participant's responses (*i.e.*, the eye for which they reported more stable binocular vision);

iv) if a participant indicated dissatisfaction with the prism correction, a lesser or greater strength was tried and prescribed;

v) for participants presenting a deviation of less than six prism diopters (PD), the prism was placed before the participant's eye of choice;

vi) for participants presenting a deviation greater than six PD, the value was divided between two prisms such that, in the absence of paresis, the prism was split equally between the two eyes; conversely, where paresis was present, the prism used before the paretic eye was twice as powerful as that placed before the dominant eye.

Thirty minutes after the prism was prescribed, the optometrist verified whether the patient continued to perceive diplopia. If diplopia was still perceived, the prism was readjusted until binocular fusion was achieved. Occasionally, a participant presented with intermittent diplopia; in these cases, the prism with the highest fusion frequency was prescribed.

At the 1-month follow-up visit, participants were required to complete the AS-20 and DQ questionnaires again. Participants that reported the prism correction as successful (diplopia rated "never" or "rarely" on the DQ for reading and straight-ahead distance) were prescribed prism eyeglasses with the same prism diopters. Participants that reported unsuccessful prism correction (diplopia rated "sometimes" or "always") went on to Phase II. During this follow-up visit, binocular vision was retested with the Worth Four Dot and the Random Dot Preschool Stereoacuity test.

### Phase II. vision therapy/orthoptics

Vision therapy/orthoptics included exercise sessions at the centre and at the participant's home. The vision therapy/orthoptics was divided into two phases: first, to improve the stability of the binocular vision and fusional vergence amplitudes; and second, to enhance the fusional vergence facility response. Table 1 shows the exercises in each phase.

The exercise sessions at the centre lasted for 2 months, with visits scheduled every 15 days (four sessions in total) and used traditional orthoptics materials and instruments such as a synoptophore, anaglyphs and vectograms, and a Brock string with and without prisms. Gamified training exercises using the previously cited virtual reality synoptophore were also used at the clinic (VisionaryVR; VisionaryTool S.L., Gijón, Spain) (*Godinez et al., 2021*). Exercises at home were prescribed at the same time, also with a 2-month duration, consisting of 20 min per day, five days per week, of game play involving computerised vergence exercises with anaglyphs (Fig. 2). Two similar programs were used: Vision Builder, to steroacutiy greater than 1,200 arc seconds (Version 2.7 for Windows; Haraldseth Software, Hamar, Norway) and VisionaryTool, to stereoacuity equal or less than 1,200 arc seconds (VisionaryTool S.L., Gijón, Spain) (Fig. 2). At each session, approximately 200 vergence responses were elicited from the participant (44 prescribed sessions ≈ 8,800 vergence responses). The Visionary Tool program is connected to the Internet and results are stored in a database hosted on a remote server. Access to this database allows the clinician to follow the participant's daily performance (horizontal and vertical vergence break/recovery value) and compliance at home.

During the follow-up visit at the end of this phase, participants were required to complete the AS-20 and DQ questionnaires again, and binocular vision was retested with the Worth Four Dot and the Random Dot Preschool Stereoacuity test.

**Table 1 Vision therapy protocol sequence.** Vision therapy protocol sequence in-office and at-home.

**PHASE ONE (first month)**

| Goals | In-office vision therapy | At-home vision therapy |
|---|---|---|
| **Enhance sensory fusion:** Increase binocular vision stability & maintenance of fusion time | **Free space activities** -Brock string **Activities with instrument** - Synoptophore | **Free space activities** -Brock string **Computer-based programs** - Games for vergence training |
| **Enhance fusional vergence ability:** Increase smooth fusional vergence amplitudes | - Vectograms - Stereoscopes **Virtual reality** | |

**PHASE TWO (Second Month)**

| | | |
|---|---|---|
| **Enhance fusional vergence facility** Increase step/jump fusional vergences | **Free space activities** - Brock string with prism bar - Brock string with loose prisms - Eccentric circles **Activities with instrument** - Synoptophore - Stereoscopes - Aperture rule - Vectograms **Virtual reality** | **Free space activities** - Brock string **Computer-based programs** - Games for vergence training |

## Data analysis

Following *Holmes et al. (2013)* criteria, successful prism treatment occurs when diplopia is rated "never" or "rarely" in both reading and straight-ahead distance gaze positions. The mean with standard deviation of the AS-20 and DQ scores were calculated for the four domains before and after prism correction, and after vision therapy/orthoptics treatment in the case of participants for whom prism correction alone had been unsuccessful.

All participants completed both questionnaires during the first visit; before prism correction; after 1 month to prism correction in the follow-up visit and 2 months later from the group that performed the vision therapy/orthoptics. AS-20 questionnaire scores before and after prism correction, and before and after vision therapy/orthoptics, were compared using a Wilcoxon signed rank test. A Mann-Whitney U test was used to analyse changes in DQ scores among participants for whom treatment had succeeded or failed, and scores pre- and post-treatment were compared using a Wilcoxon signed rank test. These two analyses were performed for the whole group, for the successful and failed treatment sub-groups, and according to the prism value. In addition, a chi-square test was performed to determine whether the success number was significant. Baseline differences between participants were analysed, considering the success or failure of both prism correction and vision therapy/orthoptics. Cronbach's alpha coefficient was used to evaluate the reliability of the scales used, despite the questionnaires having been previously validated for this purpose.

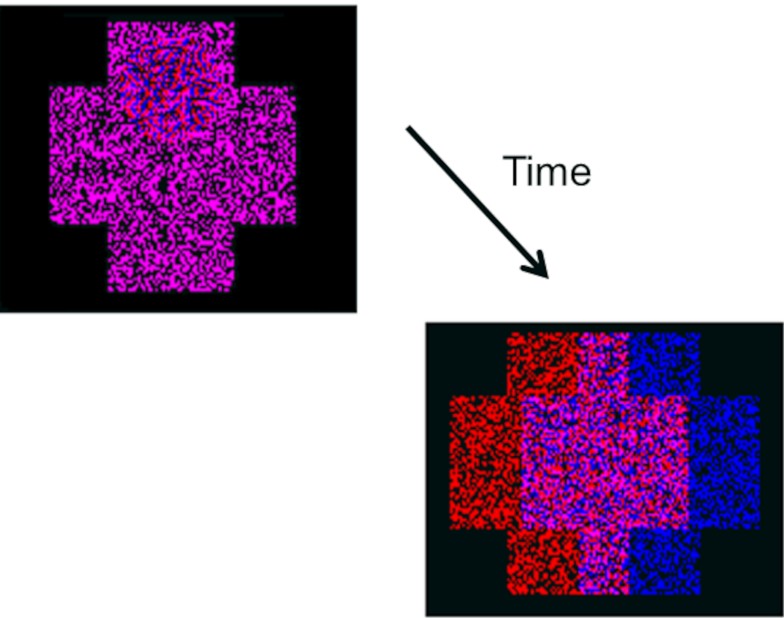

**Figure 2 Graphical example of the gamified visual therapy.** Logical process of the game. The participant's task is to locate a ball situated at one of the four extremes of a cross (top, bottom, right or left). The program automatically adjusts the vergence difficulty during the training session. Where the participant responds correctly, the software will split the image into two anaglyph crosses which the participant must then merge using their own vergence system. Should the participant fail to respond correctly, the two crosses are merged by the program.

Compliance with treatment was calculated considering that 40 sessions were prescribed to be performed with a frequency of at least five sessions per week, hence 40 vergence training sessions over a 2-month period was equivalent to 100% compliance. The following formula was used to determine percentage compliance:

$$Compliance = \frac{Sessions}{40} \times 100$$

Finally, the vergence response at home (break/recovery value) was analysed with a Wilcoxon signed rank test.

## Characteristics of the sample

A total of 48 adults (21 women and 27 men) with diplopia were recruited for the study, with a mean age of 62.45 ± 16.00 years old (within the age range 26 to 86 years). The baseline variables can be consulted in annexed 1 and 2. Sixteen of the participants had received previous treatment to correct the diplopia: nine of these had been treated with botulinum toxin; four had undergone surgery (three for strabismus and one for Graves' ophthalmopathy); and another three had been treated with botulinum toxin followed by strabismus surgery.

The spheric refraction equivalent was −1.12 ± 4.29 D (−22.00 to 4.25 D range) for the right eye and −0.94 ± 4.28 D (−22.00 to 3.50 D range) for the left. Presbyopia was compensated in 37 participants with a mean addition of 1.90 ± 1.10 D (1.50 to 3.00 D range). The median BCVA in decimal acuity was 0.92 ± 0.16 (0.40 to 1.00 range) for the

right eye and 0.92 ± 0.15 for the left (0.40 to 1.00 range). The Worth Four Dot test recorded results of diplopia at far distance in all participants except two participants, who had no strabismus at distance, achieved sensory fusion. Of the 48 participants, 32 (66.67%) were diagnosed as stereo blind at near distance; mean values were 914.60 ± 559.92" (arc seconds) within a 40" to 1,300" range.

Most of the participants presented with paresis of the 4th (15) and 6th (18) cranial nerves, four exhibited restrictive symptoms (scleral buckling for retinal detachment, myopic myopathy, and thyroid surgery), three subjects had convergence insufficiency with orthotropia at distance, two had a deviation due to decompensation in strabismus following esotropia surgery (one hypertropia and the other exotropia), and the last six presented with hypertropia or hypotropia due to overaction or underaction of the vertical muscles.

The strabismus distribution was 22 participants (45.83%) with strabismus of the right eye, and 26 (54.17%) with strabismus of the left. Twenty participants (41.67%) had a pure horizontal deviation, (with no additional vertical deviation), 17 (35.42%) had an isolated vertical deviation, and 11 (22.91%) presented a mixed deviation.

The mean value of the horizontal deviation was 5.40 ± 6.58 PD (4 to 30 PD range) at far, and 4.26 ± 8.57 PD (4 to 30 PD range) at near distance. The vertical deviation mean value was 3.75 ± 4.55 PD (4 to 20 PD range) at far, and 3.85 ± 4.66 PD (2 PD to 20 PD range) at near distance. Excyclotropia deviation was 1.55 ± 2.76 degrees (2-to-10-degree range).

## RESULTS

### Initial visit and pre-prism correction results

A total of 48 participants completed the AS20 and the DQ prior prism correction. The mean DQ score (rated from 0 to 100, best to worst) was 64.01 ± 24.46, and diplopia at far and reading distance in forty-five participants; three participants experienced diplopia only when reading. Of these, perceived diplopia was reported when looking to the right (30 participants), to the left (34), upwards (26), downwards (30), and in intermedial positions (29).

Global mean AS-20 scores before prism correction were 51.39 ± 22.52 points in the general function domain, 53.50 ± 31.80 points in the reading function domain, and 89.30 ± 15.74 points and 86.31 ± 21.24 in the self-perception and interaction domains, respectively (Fig. 3). The mean prism correction values were 7.73 ± 5.80 PD (4 to 30 PD range) at far, and 6.92 ± 7.05 PD (0 to 30 PD range) at near distance.

### Post-prism treatment overall results

Five of the 48 participants did not complete the AS20 and the DQ at the end of the prism correction period: three dropped out of the study, one died, and one declined to wear the prism for work reasons.

Based on *a priori* definition of successful and failed prism correction from participants' responses to the DQ, prism correction was classified as successful in 22 of 43 participants (51%), and unsuccessful in 21 (49%) (*p* > 0.1, according to chi-square analysis). The DQ

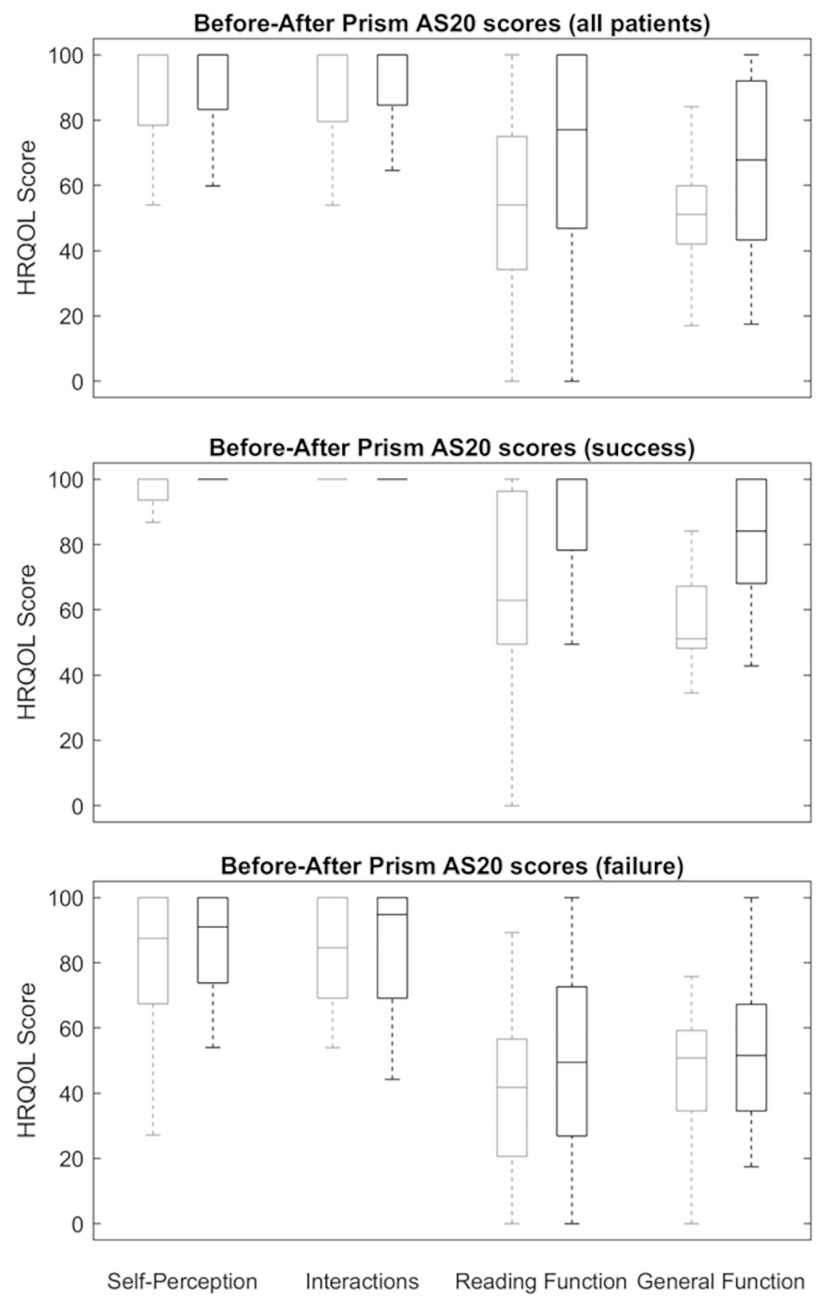

**Figure 3 Outcomes after prisma correction.** Box and whisker plots of Adult Strabismus-20 Health-Related Quality of Life scores in participants with diplopia treated with prism. Clear line boxes show pre-prism scores and shaded line boxes show scores in prism correction. The centre line represents the median, lower and upper quartiles; the whiskers represent the extremes. Top, all participants; centre, successfully treated participants only; bottom, participants who failed prism treatment.

results pre- and post-prism treatment analysed with Wilcoxon signed rank test indicated significant differences in success group ($p < 0.01$) and failure group ($p = 0.01$).

Of the 43 participants, the mean DQ score was 62.75 ± 24.67 before and 30.89 ± 32.96 after prism correction. A Mann-Whitney U test was used to analyse the effect of the prism

correction through DQ score. the differences between the successful and failed treatment groups were significant after treatment ($p > 0.01$), although DQ score before treatment did not statistically significant differences in these groups ($p = 0.73$).

Among the 43 participants, mean AS-20 scores showed a statistically significant improvement after prism correction compared to scores obtained prior to prism correction in both general function and reading function domains, improving from 51.39 ± 22.52 to 68.48 ± 24.12, and from 53.50 ± 31.80 to 68.66 ± 32.55, respectively ($p < 0.01$ in both). No significant changes in scores ($p > 0.1$ in both) were recorded in the self-perception and interaction domains (Fig. 3).

Subsequent to prism treatment rated as successful, prism eyeglasses with the same prism diopter value were prescribed. At far distance, five of the participants did not require a prism, while the mean prism power for the remaining subjects was 6.95 ± 6.13 PD, within a 2 to 30 PD range. At reading distance, 11 subjects did not require a prism, while the remainder were prescribed a mean prism power of 6.40 ± 7.66 PD, within a 3 to 30 PD range.

Before prism treatment (initial visit), of the initial 48 patients, 46 participants presented with diplopia on the Worth Four Dot test and 32 of 48 patients with null stereoacuity on the Randot Preschool Stereoacuity Test (participants with null stereoacuity, 1300" was asigned) Following prism treatment, 18 participants (37.5%) achieved fusion on the Worth Four Dot test (mean scores 4.42 ± 0.50; $p = 0.01$), and another 18 showed null stereoacuity. The overall mean score stereoacuity was (673.49 ± 567.88"; $p = 0.01$).

## Post-prism results: success or failure

Among the 22 participants for whom the prism treatment was a success, mean DQ score was 1.62 ± 3.33 (0.00 to 11.00 rate range). Mean AS-20 scores significantly improved in the reading function domain, from 65.99 ± 32.39 to 86.45 ± 23.24 points ($p = 0.01$); and in the general function domain, from 55.68 ± 22.96 to 83.32 ± 17.37 after prism correction ($p < 0.01$). No significant differences were found in either the self-perception ($p = 0.17$) or interaction domains ($p = 0.50$), as shown in Fig. 3.

Among the 21 participants for whom the prism treatment was not effective, mean DQ score was 46.69 ± 25.12 (27.70 to 100.00 rate range). Mean AS-20 scores showed no significant improvement in any domain ($p = 1.00$). These participants were older, had a greater angle of deviation, and worse binocular vision (Table 2) than success group; they also obtained a higher score on the DQ and more symptoms on the AS-20 questionnaire than success group (Table 3).

## Post-vision therapy/orthoptics overall results

Phase II (Vision therapy/orthoptics) commenced with 21 participants. One of them one of them left the study. The treatment was successful in 12 participants (60%) and failed in eight (40%). The number of participants that experienced an effective improvement was not significant ($p = 0.27$, using a chi-square test).

Mean DQ scores were 58.60 ± 22.62 before, and 20.46 ± 27.75 after vision therapy/orthoptics. A Mann-Whitney U test was used to analyse the effect of the vision therapy/

**Table 2 Comparative analysis between two groups (success _versus_ failure).**

|  | Success group N = 22 | Failure group N = 21 | _p_ |
|---|---|---|---|
| Age | 70.09 ± 13.15 | 53.85 ± 15.39 | <0.01 |
| Refraction RE | −1.50 ± 4.88 | −0.98 ± 4.09 | 0.51 |
| Refraction LE | −1.22 ± 4.88 | −1.01 ± 4.02 | 0.69 |
| Visual acuity RE | 0.88 ± 0.81 | 0.96 ± 0.8 | 0.41 |
| Visual acuity LE | 0.96 ± 0.81 | ±0.91 ± 0.169 | 0.57 |
| Worth test | 5 | 4.90 ± 0.30 | 0.14 |
| RPST | 607 ± 69 | 1219.5 ± 278.60 | <0.01 |
| Far strabismus horizontal deviation | 3.65 ± 3.43 | 8.05 ± 8.61 | 0.16 |
| Far strabismus vertical deviation | 2.64 ± 3.82 | 5.05 ± 5.39 | 0.08 |
| Near strabismus horizontal deviation | 0.91 ± 2.52 | 8.57 ± 9.25 | <0.01 |
| Near strabismus vertical deviation | 2.77 ± 3.93 | 5.43 ± 5.37 | 0.06 |
| Torsional strabismus deviation | 0.55 ± 1.26 | 2.52 ± 3.59 | 0.06 |
| Far prismatic correction | 5.85 ± 2.49 | 10.48 ± 7.59 | 0.03 |
| Near prismatic correction | 3.55 ± 4.04 | 11.33 ± 7.83 | <0.01 |

Note:
Comparative analysis between two groups (success _versus_ failure) before prismatic correction. Statistical analysis was made with the U Mann Whitney. Abbreviations: RE = Right eye; LE = Left eye; RPST = Random Dot Preschool Stereo-acuity Test. Statistical significance $p < 0.05$.

**Table 3 Comparative symptoms analysis with adult strabismus-20 test and the diplopia questionnaire.**

| Adult strabismus-20 test | Success group | Failure group | _p_ |
|---|---|---|---|
| Self-perception (0–100) | 93.23 ± 13.74 | 79.07 ± 25.33 | 0.01 |
| Interaction (0–100) | 95.12 ± 11.82 | 83.21 ± 17.24 | <0.01 |
| Reading function (0–100 | 65.99 ± 32.39 | 40.24 ± 25.90 | <0.01 |
| General function (0–100) | 55.68 ± 22.96 | 46.89 ± 21.69 | 0.18 |
| **Diplopia questionnaire test** | 56.25 ± 26.56 | 68.94 ± 21.52 | 0.07 |

Note:
Comparative symptoms analysis with Adult Strabismus-20 Test and the Diplopia Questionnaire before prismatic correction between two groups (success _vs_ failure). Statistical analysis was made with the U Mann Whitney. Statistical significance $p < 0.05$.

orthoptics. Although not significant before ($p = 0.09$), the differences between the successful and failed treatment subgroups were significant after treatment ($p > 0.01$). DQ results pre- and post-vision therapy/orthoptics analysed with Wilcoxon signed rank test exhibited significative differences ($p = 0.05$) in both group.

Over the 2-month period of the vision therapy phase, 19 participants at-home therapy compliance was calculated. One of the participants was not able to see the hidden silhouette in Visionary program.

The ranged from a maximum of 58 days to a minimum of 18. The mean compliance was 84 ± 0.21%.

Mean AS-20 scores significantly improved in the reading function domain, from 59.13 ± 29.92 to 86.93 ± 19.66 points after undergoing the vision therapy/orthoptics programme

($p < 0.01$); and in the general function domain, from $55.04 \pm 22.37$ to $75.05 \pm 20.10$ points after vision therapy/orthoptics ($p < 0.05$). No significant differences were recorded in either self-perception ($p = 0.39$) or interaction domains ($p = 0.12$), as shown in Fig. 4.

In the successful treatment subgroup, the difference in vergence values pre- and post-treatment obtained with gamified computer-based exercises at home was significantly better than in the failed treatment subgroup (Table 4).

Prior to the vision therapy/orthoptics treatment, 17 participants presented with diplopia at distance on the Worth test, and another 16 with null stereoacuity on the Randot Preschool Stereoacuity Test. After vision therapy/orthoptics, only five participants exhibited diplopia at distance on the Worth Four Dot test (mean scores: $4.25 \pm 0.44$, $p < 0.01$), four of whom also exhibited null stereoacuity (mean scores: $615.00 \pm 463.44$, $p < 0.01$).

After vision therapy/orthoptics, the prism power needed to achieve binocular vision was significantly lower at far (mean scores: $3.07 \pm 5.80$ PD; $p < 0.01$) and near distances (mean scores: $2.93 \pm 5.66$ PD, $p < 0.01$).

## Post-vision therapy/orthoptics results: success or failure

Among the 12 patients successfully treated with vision therapy/orthoptics, mean DQ score was $3.04 \pm 3.83$ (0.00 to 13.50 rate range). Mean AS-20 scores significantly improved after undergoing the vision therapy/orthoptics programme in the general function and reading function domains. General function scores improved from $55.04 \pm 22.37$ points post-prism correction to $75.05 \pm 20.10$ points subsequent to undergoing the vision therapy/orthoptics programme ($p = 0.02$). Reading function domain scores improved from $59.13 \pm 29.92$ points post-prism to $86.93 \pm 19.66$ points post-vision therapy/orthoptics ($p = 0.01$). No significant difference was found between pre- and post-vision therapy/orthoptics scores in the self-perception and interaction domains ($p = 0.89$ and $p = 0.14$, respectively), as shown in Fig. 4.

Among the eight patients for whom vision therapy/orthoptics treatment was unsuccessful, mean DQ score was $46.69 \pm 25.13$ (27.70 to 100.00 rate range). Mean AS-20 scores improved significantly in the reading function domain, from $41.83 \pm 28.42$ points post-prism correction to $52.77 \pm 28.43$ points post-vision therapy/orthoptics ($p = 0.04$). In the general function, self-perception and interaction domains, no difference was found between pre- and post-vision therapy/orthoptics scores ($p = 1.00$, $p = 0.07$ and $p = 1.00$) for respective domains, as shown in Fig. 4.

## Summary

The proposed treatment—prism correction followed by vision therapy/orthoptics as required—was successful in 34 of the 43 patients treated, or 79% of the study sample ($p < 0.01$, using a chi-square test).

In addition, the participants showed significant improvement in their binocular vision tested with the Worth Four Dot test, from $4.95 \pm 0.20$ to $4.12 \pm 0.32$ ($p < 0.01$); and in stereoacuity, from $914.60 \pm 559.92$ to $427.90 \pm 432.87$ ($p < 0.01$).

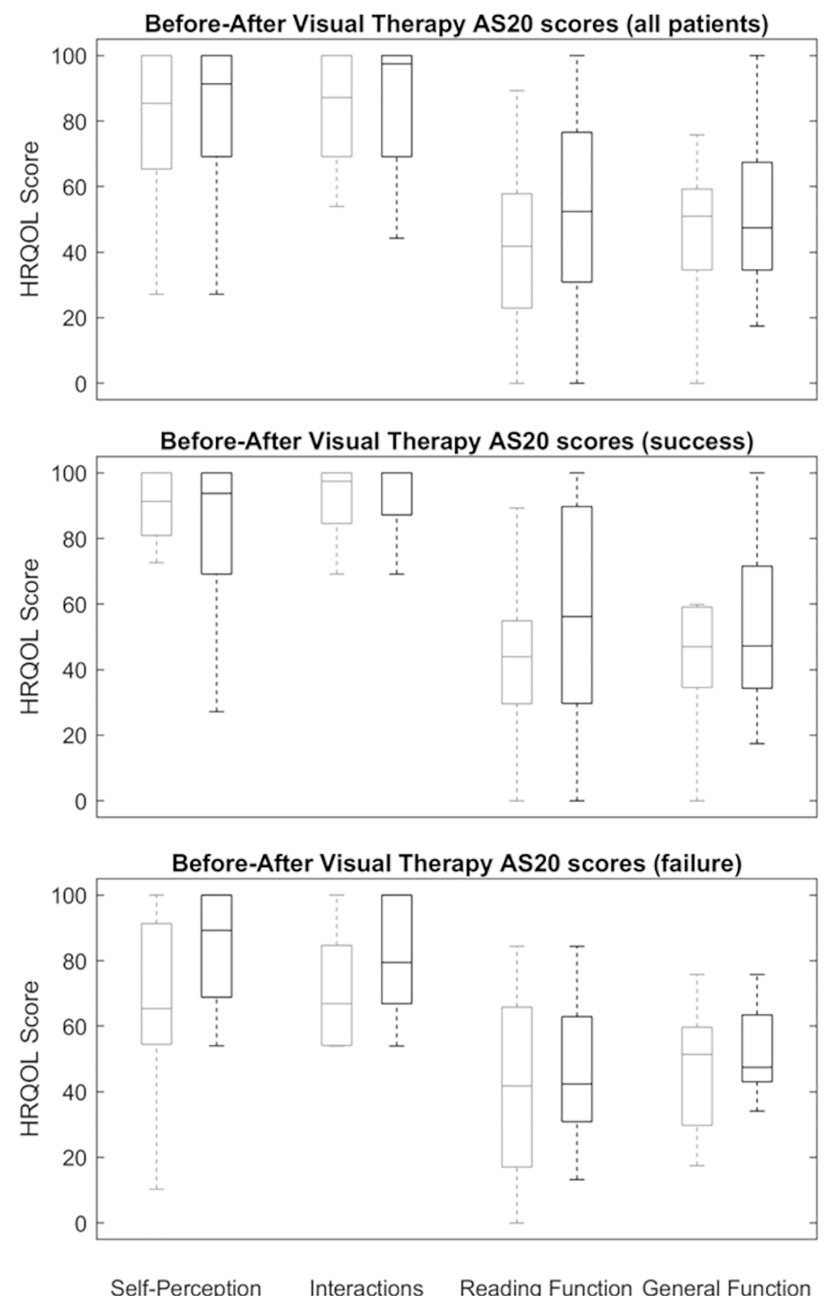

**Figure 4 Outcomes after visual therapy treatment.** Box and whisker plots illustrate Adult Strabismus-20 Health-Related Quality of Life scores in participants with diplopia treated with orthoptic/vision therapy. Clear line boxes show pre-orthoptic/vision therapy scores and shaded line boxes show scores after orthoptic/vision therapy. The centre line represents the median, lower and upper quartiles; the whiskers represent the extremes. Top, all patients; centre, successfully treated participants only; bottom, participants who failed prism treatment.

## DISCUSSION

In this study, prism correction alone resolved the diplopia in 51% of patients. When the prism correction was unsuccessful, the orthoptic/vision therapy prescribed resolved the

**Table 4 Changes in vergence depending on the group.**

|  | Success group | | | Failure group | | |
|---|---|---|---|---|---|---|
|  | Pre-treatment vergence in PD | Post treatment vergence in PD | p | Pre-treatment vergence in PD | Post treatment vergence in PD | p |
| PFV break | 21.57 ± 11.94 | 33.07 ± 17.75 | 0.14 | 17.60 ± 5.03 | 25.80 ± 15.97 | 0.14 |
| PFV recovery | 7.43 ± 4.24 | 24.07 ± 17.41 | 0.02 | 8.00 ± 4.89 | 18.60 ± 16.01 | 0.34 |
| NFV break | 9.09 ± 7.76 | 17.98 ± 8.15 | 0.01 | 9.78 ± 10.10 | 14.00 ± 13.77 | 0.03 |
| NFV recovery | 4.00 ± 3.21 | 11.48 ± 5.06 | 0.01 | 4.50 ± 3.477 | 8.43 ± 7.98 | 0.04 |
| Hyper OD break | 7.41 ± 4.50 | 9.41 ± 5.42 | 0.11 | 6.40 ± 4.08 | 7.60 ± 6.26 | 0.42 |
| Hyper OD recovery | 4.16 ± 3.12 | 6.16 ± 4.30 | 0.06 | 4.30 ± 3.23 | 5.20 ± 5.77 | 0.65 |
| Hypo OD Break | 4.33 ± 1.53 | 6.50 ± 3.68 | 0.09 | 7.20 ± 5.73 | 8.80 ± 5.62 | 0.68 |
| Hypo OD recovery | 2.66 ± 1.03 | 5.33 ± 3.55 | 0.06 | 2.50 ± 1.93 | 5.10 ± 4.10 | 0.11 |

**Note:**
Changes in the horizontal and vertical vergences response in the Successful and Failure Group. PFV, positive fusional vergences; NFV, negative fusional vergence; Hyper, hypervergences; Hipo, hypovergences. The measurements were obtained in prismatic diopters, PD. Statistical analysis was made with the Wilcoxon Test. Statistical significance $p < 0.05$.

diplopia in 60% of patients. Overall, the proposed treatment of prism correction followed by vision therapy/orthoptics as required was successful in 81% of patients.

Participants successfully treated with prisms improved in both the general function and reading function domains. These results are congruent with those obtained by *Hatt et al. (2014)* underlining the importance of prisms in the treatment of diplopia. Interestingly, those participants that failed to obtain stable fusion at far distance or when reading, according to DQ results, showed no significant improvement in either general or reading function domains. This result obtained in our study was also pointed out by *Hatt et al. (2014)*.

A Wilcoxon test showed significant before and after differences, even in the failed treatment subgroup. Participants with diplopia at near and far distances are scored with 80 (40 in straight-ahead distance plus 40 in reading), and 40 when diplopia occurs sometimes. However, a change from 80 to 40, although significant in the test, does not indicate a significant improvement in HRQOL. Successful treatment should be associated with the frequency of the diplopia, when diplopia occurs rarely or never, in straight-ahead distance or reading gaze positions, as a dichotomous variable. Analysis of the effect of prism correction and/or vision therapy/orthoptics treatment should compare DQ scores for the successful and failed treatment subgroups before and after treatment to detect possible changes. With both treatments, the differences between groups that were not significant before became significant after treatment.

The participants that went on to the vision therapy/orthoptics treatment also improved in both the general and reading function domains. Again, as with the prism correction, participants that showed no improvement on the DQ recorded no improvement on the AS-20 either. *Hatt et al. (2014)* considered that the absence of AS-20 improvement in participants classified as prism treatment failures suggests that the improvement recorded for the successfully treated participants was not attributable to a placebo effect. In this study, our findings coincide with those of *Hatt et al. (2014)* in relation to both prism treatment and visual therapy/orthoptics. However, we cannot definitively state that there has been no positive placebo effect.

Very few studies have studied the prism correction effect in adult subjects with diplopia (*Gunton & Brown, 2012*; *Tamhankar, Ying & Volpe, 2012a*, *2012b*, *2011*). *Tamhankar, Ying & Volpe (2011)* is a retrospective study in subjects with 4th cranial nerve palsy, with the prism treatment results classified into three categories: totally satisfied, mostly satisfied and dissatisfied. The analysis performed in this study, using a standardized symptom questionnaire, represents a step forward.

To the best of the our knowledge, no previous study has analysed the impact of vision therapy/orthoptics treatment on the HRQOL of subjects with diplopia. Similarities may be found in studies on the treatment of convergent insufficiency, according to which vision therapy/orthoptics have reduced the most common symptoms during reading (*Scheiman et al., 2020*). But approximately only 35% of adults with convergence insufficiency experience diplopia (*Rouse et al., 2004*). *Lijka, Toor & Arblaster (2019)* analysed speed and accuracy in subjects with binocular vision for whom double vision was caused with a vertical prism (a situation similar to a paresis of the 4th cranial nerve). The findings were reduced reading speed and accuracy compared to the control condition. Therefore, it makes sense that improvements in reading function should be obtained alongside improved binocular vision in diplopia subjects.

Baseline clinical differences between the successful and failed treatment subgroups were also analysed. Tables 2 and 3 show that participants with more intense diplopia symptoms, higher HRQOL symptomatology, worse binocular vision and higher deviation angle showed the poorest results after prism treatment. In previous studies, the strength of the prisms prescribed was not statistically associated with greater success of prism treatment (*Hatt et al., 2014*; *Tamhankar, Ying & Volpe, 2012a*). In our study, subjects with stereoacuity were significantly more likely to succeed than those subjects with null stereoacuity. It could be that vision/orthoptic therapy helps restore binocular vision and improve the prognosis of prism treatment. Another possibility could be improved vergences: subjects in the successful treatment group improved their vergence ranges significantly after vision therapy/orthoptics (Table 4), and needed lesser-powered prisms to obtain a successful score on the DQ.

Compliance with home therapy was 83.19% of the time prescribed (the mean was 41 days of home therapy). On these days, approximately 8,200 vergence responses were elicited from the participants. Vergence treatment at home ensures a high number of responses, and this in turn enhances the development of fusional vergence amplitudes. Although these were expanded in both groups, improvements in vergence response were

smaller in the failure treatment group, and insufficient to obtain successful scores (Table 4). Future studies could evaluate changes in the vergence response before and after treatment with a standardized method (*e.g.*, prism bar) to determine the impact of prism treatment on stereoacuity and vergence response.

In participants for whom prism correction was insufficient, vision therapy/orthoptics resolved the problem in 60% of cases. A clear shift towards improved HRQOL was observed, as for successful prism treatment. Sixteen of these participants had a history of strabismic surgery or botulinum toxin, rendering prism treatment and vision therapy/orthoptics the only remaining therapeutic option.

Previous studies have demonstrated the effectiveness of vision therapy/orthoptics in convergence insufficiency (*Scheiman et al., 2020*), but not, to the author's knowledge, in the case of diplopia. Although the present study sample included three participants with convergence insufficiency, only one participant showed improvement. Importantly, the two participants who showed no improvement had another condition—Parkinson's disease—that may have affected their prognosis. The prevalence of Parkinson's in adults with diplopia is high (18.1%) (*Hamedani et al., 2021*). The success of the proposed treatment in Parkinson's sufferers is therefore deserving of future study.

The present sample included participants with systemic diseases that increase the risk of depression. One of these participants, recently diagnosed with CREST syndrome, dropped out of the study. *Hatt et al. (2014)* (IOVS 2013;54: ARVO E-Abstract 5987) also describes how depression reduces self-perception of HRQOL in subjects with diplopia.

The use of new technologies and gamification strategies may contribute to increase motivation and compliance. Adherence is increased when new technologies make the treatment accessible at the patient's home as well as at the treatment centre, with remote compliance and performance tracking, as previous studies have shown in cases of convergence insufficiency (*Pediatric Eye Disease Investigator Group, 2016*).

Despite the considerable sample size of the present pre-post pseudo-experimental study, we are aware of several limitations. The first of these is the absence of a control group. When designing the study, we opted for this methodology in the interests of the participants. Other limitations were that the clinician that performed the DQ test was not masked, vergence values were not recorded pre- and post-treatment, and prism correction compliance was not recorded (on a schedule, for example). Moreover, the sample itself was very heterogeneous. Most of the participants presented with 4th and 6th cranial nerve paresis, while others exhibited different types of strabismus, such as convergence insufficiency in the geriatric population or childhood-onset esotropia. Childhood-onset esotropia usually exhibits high deviation angles, and even sensory adaptations, that make treatment and prognosis challenging, and this may have had a statistical effect on the study.

Finally, an assessment of the stability of the improvements—6 months after the end of the treatment, for example—would have been desirable.

## CONCLUSIONS

Effective prism correction of diplopia is correlated with enhanced HRQOL. Successful vision therapy/orthoptics treatment may also improve HRQOL when prism treatment alone fails to produce significant results.

## ACKNOWLEDGEMENTS

Thanks to Nerea Hernández for her help in data collection and computer support to patients.

### Funding

The authors received no funding for this work.

### Competing Interests

Santiago Martín-González and Juan Antonio Portela-Camino promoted, with the support of the University of Oviedo, the creation of the startup VisionaryTool. Both have assisted VisionaryTool, S.L. (sociedad limitada, www.visionarytool.com) to create a commercial version of vergence games program. VisionaryTool has not had any role (writing, analysis, or control over publication) in the production of the article.

### Author Contributions

- Juan Antonio Portela-Camino conceived and designed the experiments, performed the experiments, analyzed the data, prepared figures and/or tables, authored or reviewed drafts of the article, and approved the final draft.
- Irene Sanchez conceived and designed the experiments, performed the experiments, analyzed the data, prepared figures and/or tables, authored or reviewed drafts of the article, and approved the final draft.
- Cristina Gutierrez conceived and designed the experiments, performed the experiments, authored or reviewed drafts of the article, and approved the final draft.
- Santiago Martín-González conceived and designed the experiments, performed the experiments, prepared figures and/or tables, authored or reviewed drafts of the article, and approved the final draft.

### Human Ethics

The following information was supplied relating to ethical approvals (i.e., approving body and any reference numbers):

The study followed the tenets of the Declaration of Helsinki and was approved by the Basque Country Ethics Committee of Clinical Research (ref: PI2021059).

### Data Availability

The raw measurements are available in the Supplemental Files.

## Supplemental Information

Supplemental information for this article can be found online at http://dx.doi.org/10.7717/peerj.17315#supplemental-information.

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
