# Peer review of "Successful treatment of diplopia using prism correction combined with vision therapy/orthoptics improves health-related quality of life"

_PeerJ, doi:10.7717/peerj.17315_

## Round 0.1 · original submission · Major Revisions

Three reviewers have already revised your manuscript. The reviewers have found several aspects that must be clarified or changed. Please, reply in a point-by-point fashion to all these comments, and include them in the revised text. Reviewer 2 included some comments in the attached document. As explained by reviewers, there are some reference and grammatical errors, and they must be revised before further considering this manuscript for publication.

**Language Note:** The review process has identified that the English language must be improved. PeerJ can provide language editing services - please contact us at copyediting@peerj.com for pricing (be sure to provide your manuscript number and title). Alternatively, you should make your own arrangements to improve the language quality and provide details in your response letter. – PeerJ Staff

Reviewer 1 ·

Basic reporting

Abstract.
• It is not clear the explanation of the inclusion criteria, since “Inclusion criteria…. to …straight-ahead distance” The authors should better precise this sentence. It seems that the authors refer to “prism diplopia”? And this has little sense.
• The authors should explain or refer to the diplopia questionnaire. Is this a validated questionary or was a questionnaire used or developed by the authors?
• The abstract must be changed according to the comments made in the comments in each section. The sample must be explained in methods and results must only show results, no methods. Conclusions must be changed according to the results obtained. I explain. The authors have not shown if this treatment used is an option. Your results are that the therapy used obtained a particular results, so these are your conclusions.




Introduction.
• Lines 69-70. The sentence: “Researchers have proving……….reading” should be referenced.
• Line 75. You must change the sentence according to the evidence because the reference 19 in any case is all the evidence. It is better to explain that “different/several studies/authors indicate that the improvement in HRQOL…” because mentioning the reference 19 to say that this is all the evidence, it is not true. Because the reference 19 is not a systematic review with all studies in this subject but a study according to a sample examined to yield a conclusion that nondiplopic patients have more psychosocial concerns than those with diplopia, even after strabismus surgery. So please, you must change this asseveration.
• Lines 84-86. English language should be improved.




References.
• Reference 13. You have to write the adequate reference, which is published in the appropriate journal, and you have to delete the comment to “available from…”


Figures.
• Figure 3 has a typographical error. “prisma”


Tables.
• Table 1 and 2 should be specified the p value which means significance.

Experimental design

Methods.
• Line 89. According to the design of this study, it is not correct. A case series is a study in which the authors merely describe several characteristics of patients.
However, in your study you have selected a sample of 48 patients and according to your criteria, you have done a particular treatment for them. You have examined before the treatment and after it, so this is what is named a pre-post pseudo-experimental study which has more value than a case series. So please you have to change it because your study has a better scale of evidence than a case series.
• Line 105-107. The authors should explain why they have used the AS-20 questionnaire and have not used the scale which was developed using Rasch analysis. When using Rasch analysis, the authors who developed AS-20 recommended eliminating two items, scoring the questionnaire in other way. So please, you must comment this issue and explain which version you have used and why. And you should also explain or argue how this aspect could influence your results.
• Line 126-127. I understand that when saying insert model and manufacturer is an error.
• Lines 132-135. The authors should explain why they used this protocol to prescribe the prism value. Which criteria are based on? Is there any validated protocol, a reference for that?
• Line 137. The word “where” should be changed for “when”. The same in line 139, line 157, line 290… Please revise all this words because when you say where the sense is not adequate. Other example, line 222, line 290, 350, 369…
• Line 138. Why if the patients were successful a new glasses were prescribed? Was the same value of prism? It is unclear why the authors did it. They should explain this in the manuscript.
• Lines 153-166. It is not clear when is considered an improvement with the AS-20. It should be explained the score which the normal value is, to know when the authors considered that there was an improvement with this questionnaire.

Validity of the findings

Results.
• Lines 169-196. This is an important error. All this information is not a part of the section of results, but of the section of methods. Methods include the characteristics of the sample included in a study. You have selected 48 patients, and they have a mean age, different types of diagnosis, visual acuity, and so on. All of these data, belong to the section of methods because you have not done any intervention to obtain this. They are the characteristics of the patients. However, when you make your intervention, then what you yield, this is certainly the section of results.
So all of this information may be changed to the section of methods, explaining the sample. Then, the section of results must begin in the line 198.
• The authors should begin the results showing the number of patients who yielded successful and the number of unsuccess to be clear for the reader. As it is as now, it is not clear. In the lines 235-243, it seems that 22 were prism success and 21 not. So the other 5 patients were in which group? You have to improve the explanation.
• In this section of results the authors must also improve or clarify the number of patients who improved with visual therapy. It is not clear. This must be explained in a sentence to be clearer for the reader.

Discussion.
• Line 312. It should be written the authors´ because the authors are several not only one.
• Lines 314-315. The reference 29 does not refer to that. In that study what the authors do is to test how the symptoms related to convergence insufficiency vary with treatment. But it is not related to your comment about the relationship with the reading skills. In fact, there is a clinical trial w of this group of researches, which the authors have not referenced, in which it has been proven that vision therapy does not improve any aspect related to the reading abilities. So, you should change this comment.
• Line 344. Revise the sentence. It has no sense to say “the present study sample…” What do you want to explain? In any case it would be “The sample analysed…”
• Line 354. This first sentence is not true. You cannot say that this prospective study with 48 subjects makes it reliable. To know the reliability, you should have done a particular statistic. A study is not more reliable because uses a sample of patients. It depends on the type of study. So you should remove this comment and say directly the limitations of your study.
• Lines 354-358. This paragraph should be removed. You cannot say that you have not a placebo effect. In any case this is true. Always, in a type of this study, the placebo effect may be. And to comment about the ethical committee is not adequate in a study. So please remove the paragraph as it states as now. You can explain that it was considered to do this methodology because the interest of patients. But not referring to the committee or the comment to the placebo because it is not true. This is the basis of clinical trials.
Conclusions.
• Conclusions may be adequate to your results. So please limit to comment what you have encountered in your sample. That is, the prism correction that change or influence the HRQOL.

Reviewer 2 ·

Basic reporting

The writing style makes it harder to read at several places in the manuscript. I have tried to indicate that at several places but please review and consider revising where appropriate. Literature was mostly adequately cited but not sure if they were interpreted the way they were intended.

Experimental design

The experimental design needs significant revision to clearly state what the inclusion and exclusion criteria were, visit procedures for each visit etc., The criteria for prism correction needs to be explained clearly.

Validity of the findings

What was the reason for using snyoptophore when the results from that are not reported?
Why not report the motor vergence values before and after prism tx and VT?
Success is purely defined using questionnaires. Why not combine clinical measures since these questionnaires are not typically used in clinics?
The strabismus profile was 70% paretic while 60% had some vertical strabismus. How was the computer exercise alone manageable with these types of strabismus for home VT? Expanding on the types of VT activities or providing a table of VT activities used during the 3month period can enhance the applicability of the study.

Annotated reviews are not available for download in order to protect the identity of reviewers who chose to remain anonymous.

Reviewer 3 ·

Basic reporting

Well organized article. Many references included, and inclusion of raw data. I added on the PDF areas where the references don't really mention what is being said in the article after checking them. Check 22 (source 22 actually does not show an improvement in vergence ranges. please check source). 29 reference is not about reading skills but rather, performance related symptoms.

Grammatical errors found in the abstract and some throughout the article "the average" (line 32), "were" "the score" taking "into" account (line 33) , participants in the "failure group" (line 34).

Line 48: "In 81% (take out "the") of patients", then include a comma
Line 49, "additional" spelling, "enhanced" and take out "in"
Line 44: be specific about binocular fusion; do you mean sensory fusion because you use the worth four dot?

Line 54: "phoria" not "foria"
Line 75: "seem" not "seems"
Line 132, 133, 137, 139: "when" not "where"

Experimental design

There is some question on the methods:
1. How was prism amount determined? Was this consistent amongst patients?
2. Was the vision therapy protocol consistent amongst patients? What is the sequence of procedures?

Validity of the findings

Valid findings and adequate conclusions

Additional comments

did a particular subset of the population improve more with prism vs. vision therapy?
Why prism over the dominant eye?

---

## Round 0.2 · Major Revisions

I have received two revisions from the three reviewers invited to revise this version of the manuscript. I do not consider it appropriate to delay the process, and I will not wait for the third reviewer. As you can see, reviewer 2 still has some concerns about the manuscript. I agree with the reviewer that the criteria for group allocation should be carefully considered, and thus, the authors must give a scientifically sound explanation to this or modify it accordingly. The authors are encouraged to carefully address the comments of reviewer 2.

Reviewer 1 ·

Basic reporting

Authors have made appropriately the changes suggested.

Experimental design

Authors have made appropriately the changes suggested.

Validity of the findings

Authors have made appropriately the changes suggested.

Additional comments

Authors have made appropriately the changes suggested.

Reviewer 2 ·

Basic reporting

In several sections of the manuscript content was hard to follow. In particular, with the results section, stats are not reported in keeping with scholarly publications. Authors are advised to revise the results section to ensure that the content is organized

Experimental design

Thank you for providing a list of VT procedures used in this study. I am assuming the strabismus was constant in all of these participants (Annex 2). Many of them have large hyper deviations at D and or N. Did you also perform a fixation disparity test to determine if the vertical deviation was primary or secondary?
Did the authors differentiate between normal and anomalous fusion on W4D?

What specific VT activities were utilized to train vertical vergence ranges in office? Many participants had ET at D and or near with vertical deviation. Which fusional vergence was trained in each phase? What is the rationale for choosing only 2 months of VT with primary home therapy. Was correspondence determined?

Validity of the findings

Authors describe that they included a participant who declined to pursue prism treatment in the VT group. I had suggested to the authors to exclude this participant. A reanalysis should have been done, which would be the appropriate thing to do. But the authors did not follow this recommendation, rebutting that "that their inclusion provides interesting additional data to the study that may be of benefit to". This does not sound right since two of the authors are partners in the company that made the VT software.

Therefore, the current results and the manuscript should not be accepted in this state.

Annotated reviews are not available for download in order to protect the identity of reviewers who chose to remain anonymous.

---

## Round 0.3 · accepted · Accept

Reviewer 2 has not responded to my invitation to review this revision.

The authors have addressed the remaining comments, and thus, I consider that this manuscript can be accepted for publication.